# The Salt Tolerance–Related Protein (STRP) Is a Positive Regulator of the Response to Salt Stress in *Arabidopsis thaliana*

**DOI:** 10.3390/plants12081704

**Published:** 2023-04-20

**Authors:** Anna Fiorillo, Michela Manai, Sabina Visconti, Lorenzo Camoni

**Affiliations:** 1Department of Biology, University of Rome Tor Vergata, 00133 Rome, Italy; anna.fiorillo@uniroma2.it (A.F.); michela.manai@uniroma2.it (M.M.); 2Ph.D. Program in Cellular and Molecular Biology, Department of Biology, University of Rome Tor Vergata, 00133 Rome, Italy

**Keywords:** saline stress, intrinsically disordered proteins, abscisic acid, oxidative stress, abiotic stress

## Abstract

Salt stress is a major abiotic stress limiting plant survival and crop productivity. Plant adaptation to salt stress involves complex responses, including changes in gene expression, regulation of hormone signaling, and production of stress-responsive proteins. The Salt Tolerance–Related Protein (STRP) has been recently characterized as a Late Embryogenesis Abundant (LEA)–like, intrinsically disordered protein involved in plant responses to cold stress. In addition, STRP has been proposed as a mediator of salt stress response in *Arabidopsis thaliana*, but its role has still to be fully clarified. Here, we investigated the role of STRP in salt stress responses in *A. thaliana*. The protein rapidly accumulates under salt stress due to a reduction of proteasome–mediated degradation. Physiological and biochemical responses of the *strp* mutant and *STRP*–overexpressing (*STRP* OE) plants demonstrate that salt stress impairs seed germination and seedling development more markedly in the *strp* mutant than in *A. thaliana wild type* (*wt*). At the same time, the inhibitory effect is significantly reduced in *STRP* OE plants. Moreover, the *strp* mutant has a lower ability to counteract oxidative stress, cannot accumulate the osmocompatible solute proline, and does not increase abscisic acid (ABA) levels in response to salinity stress. Accordingly, the opposite effect was observed in *STRP* OE plants. Overall, obtained results suggest that STRP performs its protective functions by reducing the oxidative burst induced by salt stress, and plays a role in the osmotic adjustment mechanisms required to preserve cellular homeostasis. These findings propose STRP as a critical component of the response mechanisms to saline stress in *A. thaliana*.

## 1. Introduction

Excessive soil salinity is a dangerous environmental condition for plants [1]. The ionic and osmotic imbalance induced by high salinity impairs water and nutrient uptake and alters seed germination, flowering, and fruiting ability [2,3]. Under salt stress, plants lose their photosynthetic capacity, affecting the availability of resources required to sustain plant growth [4,5]. To overcome the adverse effects of salt stress, plants evolved several mechanisms, including vacuole compartmentalization of toxic ions, activation of antioxidant defenses, regulation of hormone levels, and accumulation of stress-responsive proteins [4].

The Salt Tolerance–Related Protein (STRP) is a 16 kDa highly hydrophilic protein of *Arabidopsis thaliana* [6,7]. The amino acid sequence of STRP is 59% identical to ST6–66, a *Thellungiella halophila* protein that confers salt tolerance when overexpressed in *A. thaliana* [6]. Moreover, the *strp* loss of function mutant is hypersensitive to salt stress; hence, the name Salt Tolerance–Related Protein was proposed [6].

Despite its self-explanatory name, the function played by STRP in salt stress remains largely unknown. In addition to the similarity with ST6–66, the only indication that demonstrates the involvement of STRP in the salt stress response mechanisms is the reduced germination rate of the *strp* mutant in the presence of NaCl [6]. Another piece of evidence suggests the involvement of STRP also in temperature stress responses in *A. thaliana*. In fact, short-term exposure to heat (42 °C) and cold (4 °C) stress increases protein levels by approximately an order of magnitude [8].

The amino acid sequence of STRP also has a 42% identity with that of WCI16, a wheat protein involved in freezing tolerance, sharing common features with Late Embryogenesis Abundant (LEA) proteins, a group of multifunctional, hydrophilic stress-related proteins [9]. Primary structure analysis has established that STRP is a highly hydrophilic protein with extensive intrinsically disordered domains and with small regions that may have a defined secondary structure. Intrinsically disordered proteins (IDPs) are involved in many cellular processes, including stress responses. Indeed, the lack of a defined structure allows these proteins access to functions inaccessible to well-folded proteins because of their unique structures [10,11]. Although STRP cannot be classified as a canonical LEA protein based on its amino acid sequence, it shares some properties with LEA proteins, such as cryoprotectant and anti-Reactive Oxygen Species (ROS) activity [7]. STRP localizes in different cellular compartments, such as the plasma membrane, cytosol, and nucleus. Cold stress triggers STRP dissociation from the plasma membrane, suggesting a possible role for the protein as a stress sensor [7]. In contrast, protein levels strongly increase in the cytosol and the nucleus due to a mechanism involving protein stabilization [7,8].

In the nucleus, STRP has been reported to interact with DEK3, a nucleosome–binding protein of *A. thaliana* that regulates chromatin accessibility and modulates the expression of target genes [12,13]. DEK3 acts as a negative regulator of salt stress responses by inhibiting the expression of stress-related genes [13]. Hence, the interaction of STRP with DEK3 strengthens the hypothesis that this protein can play a role in salt stress responses.

The STRP levels are increased by abscisic acid (ABA), which is known to play a role in the plant response to abiotic stress, including salt stress. The hormone acts by activating STRP expression and a post-translational mechanism stabilizing the protein [7]. In turn, the *strp* knockout mutant shows a significant hyposensitivity to the hormone and reduced expression of Nced3, the key rate-limiting enzyme in ABA biosynthesis in response to cold stress and exogenous ABA application, thus suggesting the involvement of STRP in ABA signaling [7].

With this background, this work aimed to deeply investigate the role of STRP in *A. thaliana* salt stress responses. For this purpose, we first investigated the effects of NaCl treatment on *STRP* gene expression and protein levels. Further, we generated and characterized *STRP*–overexpressing (*STRP* OE) lines, along with the *strp* knockout mutant, to clarify the impact of STRP on different physiological and biochemical responses under salt stress. Together, our findings have identified STRP as an important mediator of the responses to salt stress in *A. thaliana*.

## 2. Results

### 2.1. Analysis of STRP Levels and STRP Expression under Salt Stress

To test the effects of salt stress on STRP protein levels, *A. thaliana* seedlings were grown for two weeks in Murashige and Skoog (MS) medium and then transferred to MS medium supplemented with NaCl ranging from 50 mM to 150 mM. After 18 h of salt exposure, total proteins were extracted and analyzed in a Western blot using an α–STRP antibody [7]. As shown in Figure 1A, all NaCl concentrations tested induced a substantial increase in STRP protein levels. Time-course experiment demonstrated that the increase of STRP in response to salt stress was very fast, reaching the maximum after 2 h of salt exposure (Figure 1B). To ascertain whether STRP accumulation was due to gene transcription, *STRP* transcript levels were analyzed by RT-qPCR in seedlings subjected to salt stress. As shown in Figure 1C, 150 mM NaCl treatment did not affect *STRP* mRNA levels, thus suggesting the involvement of a post-translational mechanism raising STRP protein levels. In this regard, it has been recently shown that cold stress increases STRP levels by inhibiting the protein degradation mediated by the proteasome [7]. To investigate whether a similar mechanism occurred under salt stress conditions, *A. thaliana* seedlings grown in MS medium were transferred on MS supplemented with 150 mM NaCl, containing the proteasome inhibitor MG132, and kept in the growth chamber for 18 h. As shown in Figure 1D, STRP levels strongly increased in the presence of MG132 in the unstressed sample, reaching similar levels to those observed under salt stress conditions. These results demonstrate that, in control conditions, STRP is targeted for the 26S proteasome–dependent degradation and that salt stress triggers a mechanism leading to protein stabilization through proteasome inhibition.

### 2.2. STRP Overexpression

To investigate the role of STRP under salt stress, transgenic *A. thaliana* plants overexpressing STRP–YFP were produced by *Agrobacterium*–mediated transformation. Three different overexpressing (OE) lines (i.e., OE1, OE2, and OE3) were selected for kanamycin resistance; the presence of transgenic construct was confirmed by PCR amplification, using specific primers for the 35S promoter (Appendix A) and by fluorescence confocal microscopy, that showed STRP localized at the level of the plasma membrane, cytosol, and nucleus (Appendix A) [7]. The RT-qPCR analysis demonstrated a substantial increase of *STRP* transcripts in OE1, OE2, and OE3 T2 transgenic *Arabidopsis* lines (Figure 2A). Accordingly, immunodecoration with anti-YFP antibodies showed that STRP–YFP abundance in the total protein extracts was similar in the three lines (Figure 2B, left panel); moreover, staining with anti-STRP antibodies demonstrated that the protein was highly expressed compared to endogenous STRP (Figure 2C, right panel).

### 2.3. Physiological Responses to Salt Stress of the Strp Mutant and STRP OE Plants

To investigate the role of STRP in salt stress, the responses of the *strp* knock–out mutant and *STRP* OE lines to high salinity stress were studied. The mutant is homozygous for a T–DNA insertion into the promoter [14], thus preventing *STRP* gene expression [7]. The *STRP* overexpressing OE1 line (hereinafter *STRP* OE) was used for these analyses. First, the germination rate under salt stress conditions was evaluated. To this purpose, seeds were sown on MS plates supplemented with different NaCl concentrations. After four days of stratification in the dark at 4 °C, the plates were transferred to the growth chamber, and after one week, the number of germinated seeds was determined. As shown in Figure 3A, in control conditions, *strp* and *STRP* OE seeds have a germination rate comparable to *A. thaliana wild type* (*wt*) seeds. As expected, 100 and 150 mM NaCl reduced the germination rate of *wt* plants. However, for both concentrations, the inhibitory effect on germination was significantly increased in the *strp* mutant and strongly reduced in *STRP* OE seeds.

The effect of high salinity on the growth of *strp* and *STRP* OE seedlings was also investigated. Four-day-old seedlings were transferred onto MS plates containing different NaCl concentrations and grown vertically for seven days. Interestingly, leaf area was significantly reduced in the *strp* mutant and increased in *STRP* OE plants, compared to the *wt*, at every NaCl concentration tested (Figure 3B). Similarly, the roots of the *strp* mutant were significantly shorter than *wt* roots at all NaCl concentrations (Figure 3C), while the root length of *STRP* OE plants was significantly increased, compared to the *wt*, at 100 mM and 150 mM NaCl. Overall, these results suggest that *STRP* plays a role in mitigating the inhibitory effect of salt stress on seed germination and plant growth.

### 2.4. Biochemical Responses to Salt Stress in the Strp Mutant and STRP OE Plants

Chlorophyll (Chl) depletion is a well–described process in many plants under salt stress conditions [15]. Hence, levels of total Chl, Chl a, and Chl b were determined spectrophotometrically in leaves of three-week-old *strp* and *STRP* OE plants. As shown in Figure 4A, salt stress substantially reduces total Chl, Chl a, and Chl b in the three plants’ genotypes. This effect was particularly severe in the *strp* mutant, where a reduction of 34.6% of Chl a, 38.2% of Chl b, and 33.8% of total chlorophyll was observed compared to *wt* plants exposed to the same conditions. By contrast, salt treated *STRP* OE plants had higher chlorophyll levels than *wt* and *strp* plants. Indeed, under salt stress, these plants had 62.9% Chl a, 70% Chl b, and 67.2% total chlorophyll more than *wt* and 77.2% Chl a, 81.4% Chl b, and 78.3% total chlorophyll more than *strp* plants.

Chlorophyll depletion is generally considered a typical manifestation of oxidative stress conditions [16]. Reactive oxygen species (ROS), such as H_2_O_2_, HO•, O^−^_2,_ and ^1^O_2_, are overproduced under salt stress due to the ionic imbalance induced by high salinity [17,18]. To understand if STRP also has a role in preventing ROS accumulation, in situ H_2_O_2_ detection with the 3,3′–diaminobenzidine (DAB) chromogen was performed [19]. DAB is oxidized by H_2_O_2_ produced in plants, with a reaction catalyzed by the endogenous peroxidase, forming a brownish insoluble precipitate. Figure 4B shows the DAB staining of two-week-old *A. thaliana wt, strp,* and *STRP* OE leaves, treated for 18 h with 150 mM NaCl. In non-stress conditions, no H_2_O_2_ production occurs in the three genotypes (upper panel). Under salt stress (lower panel), DAB staining showed a significant H_2_O_2_ production in *wt* and *strp* leaves, with a dye intensity higher in the mutant than *wt*, indicating that, under salt stress, *strp* plants accumulate more H_2_O_2_ than *wt*. Interestingly, in the same stress conditions, no precipitate was revealed for *STRP* OE leaves, suggesting a role for STRP in regulating ROS production and accumulation. 

Plant protection against oxidative stress involves the inhibition of ROS production and the activation of enzymatic and non-enzymatic scavenging mechanisms [17]. Among these processes, the reaction catalyzed by ascorbate peroxidase (APX) contributes to detoxifying plant cells from H_2_O_2_ [17]. APX plays a central role in ROS scavenging in cytosol, mitochondria, and peroxisomes under salt stress [20]. To investigate the effect of salt stress on APX activity, whole cellular extracts of *wt, strp,* and *STRP* OE plants were used. As shown in Figure 4C, no difference in APX activity in the three lines was observed. Salt stress slightly increased APX activity in the *strp* mutant, while a significant activation of the enzyme in *wt* and *STRP* OE genotypes was observed. This activation was higher in the plants overexpressing *STRP* compared to *wt,* thus suggesting that STRP plays a relevant role in strengthening plant responses to salt–induced oxidative stress.

ROS accumulation leads to severe oxidative damage to the main cellular structures, such as membranes, proteins, and nucleic acids [21,22]. Considering this, the malondialdehyde (MDA) content and the relative electrolyte conductivity (REC) were evaluated in two-week-old *wt*, *strp*, and *STRP* OE plants treated for 18 h with 150 mM NaCl. MDA is an oxidation product of unsaturated fatty acids, while REC reveals the membrane permeability to ions. As shown in Figure 4D, salt stress induces MDA production in *wt* and *strp* plants, and this increase was higher in the mutant than in *wt*. The accumulation of MDA was inhibited in *STRP* OE plants, where no differences between control and salt stress conditions were observed. Likewise, high salinity significantly increases ion leakage in the three genotypes (Figure 4E). This increase was higher in the mutant than *wt* and strongly reduced in *STRP* OE compared to *wt* and *strp* plants. Interestingly, without stress, *STRP* OE plants had lower REC levels than *wt* plants and the *strp* mutant, suggesting that STRP has a role in reducing basal membrane oxidative damage.

Osmotic stress and sodium ions toxicity are the main effects of salt stress in plants [23]. To counteract the osmotic imbalance and the oxidative damage induced by high salinity conditions, plants have evolved different mechanisms, such as accumulating osmo-compatible solutes, mainly proline, in the cytosol. Since proline accumulation under salt stress correlates to stress tolerance [24], we compared proline production on *wt, strp,* and *STRP* OE plants treated with 150 mM NaCl. As shown in Figure 4F, no significant difference in proline content among the three plant lines was observed in physiological conditions. Under salt stress, the proline content of the *strp* mutant remains comparable to the control-untreated condition, suggesting an impairment of these plants to produce proline as a response to high salinity. On the other hand, both *wt* and *STRP* OE plants increased the proline accumulation, and this effect was significantly higher in the transgenic plants than in *wt.*

### 2.5. ABA Accumulation and NCED3 Expression under Salt Stress in wt, strp, and STRP OE Plants

Abscisic acid (ABA) plays a pivotal role in salt stress responses [25]. Salt stress strongly increases ABA levels, determining the activation of many stress-related gene expressions in an ABA–dependent manner [26]. ABA strongly upregulates *STRP* gene expression and protein levels, and the *strp* mutant has an impaired response to the hormone [7]. Hence, the accumulation of ABA under salt stress was studied in the *strp* mutant and *STRP* OE plants. As shown in Figure 5A, in the absence of stress, ABA levels are comparable in *wt, strp,* and *STRP* OE plants. Under salt stress, in the *strp* mutant, no increase in the ABA content was observed. On the contrary, salt treatment induces the accumulation of ABA in *wt* and *STRP* OE plants, with substantially higher levels in the transgenic plants.

Endogenous ABA levels are finely regulated by multiple developmental and environmental factors [27]. The 9-cis-epoxycarotenoid dioxygenase (NCED) is the enzyme catalyzing the rate-limiting step of the ABA biosynthetic pathway [28,29]. To understand if the ABA levels detected in the three genotypes under salt stress depend on a different expression of *NCED3*, an RTqPCR analysis was performed. As shown in Figure 5B, no difference in the *NCED3* expression under control conditions for *wt, strp,* and *STRP* OE plants was observed. Salt stress differently upregulates *NCED3* expression in the three genotypes; indeed, the gene was less activated in the *strp* mutant compared to *wt* and *STRP* OE plants, where higher levels of *NCED3* mRNA were detected. Hence, this finding demonstrated that STRP regulates ABA levels in response to salt stress.

## 3. Discussion

Current knowledge describes the relevant role of STRP in cold stress responses [7]. Less is known for salt stress. The high similarity of STRP with the *T. halophila* salt stress-responsive protein ST6–66 and the reduced germination of *strp* mutant under salt stress [6] prompted us to investigate the role of the protein in saline stress in depth.

Salt stress responses start from perceiving external stimuli, e.g., the ionic and the osmotic imbalance [30,31], activating complex stress response mechanisms [5]. Our results demonstrated that salt stress rapidly raises STRP levels, suggesting a function for the protein in the early responses to salt stress. Under salt stress, a notable increase in soluble stress-related protein occurs due to gene expression activation and post–transcriptional or post–translational mechanisms [32,33,34]. Our data revealed no transcriptional activation of the *STRP* gene under salt stress, while protein stability was increased. Indeed, MG132 treatment brought about STRP accumulation in non-stressed plants, reaching comparable levels of salt–treated samples. This finding demonstrates that STRP is targeted for proteasome degradation in physiological conditions, i.e., without stress., thus suggesting that the stabilization of the protein is required for plant adaptation to high salinity. Similar mechanisms have already been described in plants [35,36]. The transcription factor Dehydration–Responsive Element Binding Protein 2A (DREB2A), which regulates the expression of several stress-responsive genes [37], is unstable under non-stress conditions, and heat stress and drought stabilize the protein by blocking its ubiquitination [38].

The phenotypical analysis of the *strp* mutant and *STRP* OE plants highlighted the protective role of STRP under salt stress. High salinity affects many aspects of *A. thaliana* development, including seed germination, and root and shoot growth [39]. Results showed that the adverse effects of salt were more severe in the *strp* mutant compared to the *wt*. Notably, *STRP* overexpression significantly suppresses the deleterious effects of saline stress on germination and seedling growth, demonstrating the ability of STRP to counteract the osmotic imbalance and the ion toxicity induced by salt stress. The protective effect of STRP is very likely ascribed to its biochemical properties. In fact, IDPs are typically involved in seed germination and in the regulation of early stages of plant development by sensing unfavorable environmental conditions and consequently protecting macromolecules and intracellular structures [31,40].

A typical consequence of saline stress is a significant reduction of leaf chlorophyll content due to damage to the photosynthetic apparatus [41]. Notably, chlorophyll decreased more severely in the *strp* mutant, while *STRP* OE plants maintained higher chlorophyll levels than *wt* plants. The depletion of chlorophyll strictly correlates to oxidative burst, which, under salt stress, occurs predominantly in the form of H_2_O_2_ [42]. Interestingly, H_2_O_2_ production and ROS–induced lipid peroxidation were higher in the *strp* mutant and hampered in *STRP* OE plants. Hence, the overall evidence suggests that STRP exerts its protective function by counteracting the oxidative burst induced by salt stress. The activation of APX is an important mechanism allowing ROS detoxification under high salinity stress. In fact, APX overexpression in *A. thaliana* strongly increases the tolerance to salt stress [43]. In this context, it is worth noting that *STRP* overexpression strongly increased APX activity under saline stress. This could be due, at least partially, by the ability of STRP to preserve the enzymatic machinery responsible for ROS scavenging. Obtained data also showed that STRP plays a role in the osmotic adjustment that, in high salinity, is required to maintain cellular homeostasis. Proline is the main compatible solute accumulated to counteract the stress-induced ionic and osmotic imbalance [44]. Since it is well-accepted that proline accumulation correlates with plant salt tolerance, the finding that proline levels were reduced in the *strp* mutant and increased in *STRP*–OE plants demonstrates the ability of STRP to participate in this adaptation mechanism.

The stress hormone ABA plays an intricate role in salt stress, regulating the expression of various stress-related genes involved in synthesizing osmo–protectants and tolerance proteins [45,46].

The inability of *strp* to raise the hormone’s levels in stress conditions and, as expected, the high accumulation in STRP–OE plants provide strong evidence for the involvement of STRP in the mechanisms regulating the ABA increase in response to salt stress. This hypothesis is supported by the correlation that we observed between STRP levels and the *NCED3* gene expression. NCED3 catalyzes the primary rate-limiting reaction of the ABA biosynthetic pathway [47]. Interestingly, the expression of *NCED3* was inhibited in the *strp* mutant and up-regulated in *STRP*–OE plants, demonstrating the importance of STRP in the mechanisms controlling the production of ABA under salt stress. Since high concentrations of ABA can activate *STRP* expression [7], obtained results suggest the existence of a circuit where ABA promotes the synthesis of STRP, which is necessary for the increase of the hormone upon salt stress.

Different hypotheses can be formulated regarding the possible mechanism underlying the protective role of STRP in response to saline stress. STRP is a hyper–hydrophilic protein without any predicted secondary structure. However, it has been proposed that some protein regions may assume a conformation in a context–dependent manner, allowing a specific interaction with target proteins [7]. In this respect, it is noteworthy the identification of STRP as an interactor of DEK3. This nucleosome–binding protein regulates chromatin remodeling, and consequently, the expression of salt stress-related genes [12,13]. The interaction with DEK3 or other nuclear proteins could be the basis of the observed protective effect. Alternatively, STRP could act through a less specific mechanism, typically played by proteins that share with STRP the presence of large unstructured regions, such as LEA proteins and other intrinsically disordered proteins [48]. Proteins bearing these disordered domains typically act as chaperones, allowing the maintenance of the correct folding of their targets, and consequently, preserving their activity. In this context, the protective role of STRP could depend on its direct interaction with the protein machinery responsible for redox homeostasis in salt stress conditions. In conclusion, the results of this work demonstrate the protective role of STRP in plant adaptation to salt stress. The stabilization of the protein in response to high salinity is a mechanism that allows limiting the damage induced by oxidative stress, thus reducing alterations of basic physiological processes. STRP also acts by regulating ABA production, providing an additional level of plant adaptation to the stress condition. Hence, this work provides another piece of knowledge about the involvement of STRP in abiotic stress tolerance mechanisms.

## 4. Materials and Methods

### 4.1. Plant Material

*A. thaliana wild type* (*wt*), *strp* knockout mutant, and transgenic *STRP* OE plants used in this work belong to the Columbia–0 (Col–0) ecotype. The *strp* mutant (SALK_076125) was provided by the Nottingham Arabidopsis Stock Center (NASC).

### 4.2. Vector Construction and Agrobacterium–Mediated Transformation of A. thaliana Plants

The *STRP* coding sequence (*At1g13930*) was cloned upstream of the *YFP* sequence in a modified pGreen 0029 binary vector containing a double 35S promoter and the translational enhancer sequence of TEV [49]. The STRP transcript was obtained by RT-PCR using total RNA extracted from *A. thaliana* seedlings and the following primers: Fw 5′–CATGCCATGGCAATGAATTTCATCTCCGATCAGG–3′ Rev 5′–CATGCCATGGGCTTCAAGAAACCTTGAGCCATC–3′ containing the *Nco*I site (underlined) to allow the insertion of the amplified fragment at the 5′ end of the *YFP* sequence. The *A. thaliana* total RNA was extracted as described by [50], and reverse transcribed with the PrimeScript™ RT Reagent Kit with gDNA Eraser (Takara Bio Inc., Kusatsu, Japan) following the manufacturers’ recommendations. The PCR–amplified cDNA was controlled by DNA sequencing (Eurofins Genomics, Ebersberg, Germany) before proceeding to *Agrobacterium* transformation. Modified pGreen0029 binary vector was introduced through electroporation in the GV3101 Agrobacterium strain, which harbors the pSOUP vector, using a Micropulser Electroporation apparatus (Bio-Rad, Hercules, CA, USA).

*A. thaliana* was then transformed with the Floral Dip method [51], and the transgenic lines were selected on 50 μg/mL kanamycin. The overexpression of the construct was assessed by Western blot and RT-qPCR.

### 4.3. Growth Conditions and Salt Stress Treatment

For salt stress and MG132 treatments, *A. thaliana wt* plants were grown in MS medium (Duchefa, Haarlem, The Netherlands) containing 0.05% MES, 1% sucrose, and 0.8% agar, pH 5.7. Briefly, *A. thaliana wt*, *strp* mutant, and *STRP* OE seeds were surface sterilized in 70% ethanol and 1% sodium hypochlorite solution containing 0.05% Tween–20, washed with sterile water, and then sown on MS medium. After three days of stratification at 4 °C in the dark, seeds were transferred into a growth chamber at 22 °C, 80% humidity, under a 16/8 h light/dark cycle. After two weeks, seedlings were transferred from control media to MS supplemented with 100 µM MG132 or different NaCl concentrations for times varying according to the experiment [7].

For chlorophyll content, *A. thaliana wt*, *strp* mutant, and *STRP* OE plants were grown in soil at 22 °C, 80% humidity, under a 16/8 h light/dark cycle for one month and then irrigated with a solution of 150 mM NaCl every 3 days for two weeks.

### 4.4. Germination Assay, Primary Root Elongation, and Leaf Area Measurement

To assess the effect of NaCl on germination, two hundred seeds *wt*, *strp,* and *STRP* OE were sown on MS plates supplemented with 100 and 150 mM NaCl, stratified four days in the dark, and then moved to the growth chamber. The percentage of germinated seeds was determined after one week [6].

For root elongation and leaf area measurement, two hundred seeds *wt*, *strp,* and *STRP* OE were sown and germinated on MS medium. After four days, seedlings were transferred to MS medium supplemented with 50, 100, and 150 mM NaCl for additional seven days. The inhibition of primary root elongation under salt stress was determined by analyzing digital pictures with the ImageJ software. Leaf area was measured by analyzing digital images with Easy Leaf Area software [52].

### 4.5. Whole Cellular Extract Preparation

The whole cellular extract was prepared by grinding one g of *wt*, *strp,* and *STRP* OE seedlings with liquid nitrogen. Samples were resuspended in 2.5 mL of extraction buffer (25 mM Tris–HCl, 1 mM EDTA, 150 mM NaCl, 10% glycerol, 5 mM DTT, 2 mM PMSF, 0.5 mM ascorbic acid, 2% polyvinylpyrrolidone, 0.1% Nonidet P–40, pH 7.5) and centrifuged at 8000× *g* for 30 min at 4 °C. Supernatants were collected and protein concentration determined with the Bradford method [53], using BSA as the standard.

### 4.6. Western Blot

Proteins were separated on SDS–Page in a Mini Protean apparatus (Bio-Rad, Hercules, CA, USA) [54] and then electroblotted on PVDF membrane with the Trans–Blot Turbo Transfer System (Bio-Rad, Hercules, CA, USA), according to the manufacturer’s instructions. Anti-STRP [7] 1:10,000, anti-actin (Agrisera, Vännäs, Sweden) 1:5000, and anti-GFP (Santa Cruz Biotechnology, Inc., Heidelberg, Germany, DE) 1:1000 dilutions were used. Protein detections were performed by incubating the membrane with horseradish peroxidase–conjugated anti-rabbit secondary antibody (1:5000) or anti-mouse secondary antibody (1:5000) from Bio-Rad.

### 4.7. Ascorbate Peroxidase Activity Assay

APX activity was measured on *wt*, *strp,* and *STRP* OE whole cellular extract immediately after adding 10 mM H_2_O_2_ to a reaction mix composed of 120 µg of whole cellular extract, 3 mM EDTA, 0.5 mM ascorbic acid and 25 mM Tris–HCl, pH 7.5 with a UV–Vis Spectrophotometer (DU530, Beckman Coulter, Brea, CA, USA) at 290 nm for 5 min. APX units were calculated as follows:$$\mathrm{APX}\;\mathrm{unit}=\frac{V\cdot \Delta APX\cdot 1000}{\varepsilon \cdot l\cdot t\cdot v}$$

APX units = (ΔA_290_: change in the absorbance/min; *V*: total volume of reaction mixture; *v*: volume of sample; *ε*: extinction coefficient of ascorbate 2.8 mM^−1^ cm^−1^; *t*: time of reaction; *l*: path length). The APX–specific activity was calculated by normalizing the values obtained with the mg of sample in the reaction mix.

### 4.8. Gene Expression Analysis

Gene expression analysis was performed by RTqPCR on total RNA obtained as described above by using QuantStudio™ 3 Real–Time PCR System (Applied Biosystems™, Waltham, MA, USA), the PowerUp™ SYBR™ Green Master Mix (Applied Biosystems™), and the following specific primers (Sigma-Aldrich Corporation, St. Louis, MO, USA): *STRP* Fw 5′ TGGCAAGTGCCAAGGTTGTA 3′ Rev 5′ TACCGTATTTCTCGGCAGCG 3′; *NCED3* Fw 5′ TGTCCTGTCTGAAATCCGCC 3′ Rev 5′ CCCTGCTTCGAGGTTGACTT 3′; actin (*ACT 8*) Fw 5′ TCAGCACTTTCCAGCAGATG 3′ Rev 5′ ATGCCTGGACCTGCTTCAT 3′. Gene–of–interest values were normalized with a factor based on the two most stably expressed housekeeping genes out of *ACT8*, *UBQ10*, and *EF1α*, with the lowest M-value (<0.5). Normalized mRNA expression was calculated for each sample with the 2 ^(ctmin−ctx)^ method [55].

### 4.9. Lipid Peroxidation and Relative Electrolyte Conductivity

Lipid peroxidation was measured by MDA quantification, according to [56]. One hundred mg of two-week-old *A. thaliana wt*, *strp* mutant, and *STRP* OE plants were grounded with liquid nitrogen and incubated with 500 µL of 0.1% trichloroacetic acid (TCA). Samples were centrifuged for 10 min at 15,000× *g* at 4 °C to remove the extracellular debris, and the supernatant was added to 1.5 mL of 20% TCA containing 0.5% thiobarbituric acid (TBA). After 30 min of incubation at 100 °C, MDA concentration was determined by measuring the absorbance at 532 and 600 nm with a spectrophotometer (SmartSpec 3000, Bio-Rad, Hercules, CA, USA). Purified MDA was used as a standard (Sigma-Aldrich Corporation).

The relative electrolyte conductivity (REC) test was performed according to [57]. Briefly, 0.2 g of *A. thaliana wt*, *strp* mutant, and *STRP* OE leaf strips (5 mm) were submerged in 5 mL of deionized water, incubated for 2 h at RT, and the electrical conductivity of the solution determined with an electrical conductivity meter. Boiled samples were used to determine the maximum percentage of electrolyte leakage.

### 4.10. Determination of Free Proline

Free proline content was determined according to [49]. Briefly, 0.5 g of two-week-old *A. thaliana wt*, *strp* mutant, and *STRP* OE plants were homogenized in liquid nitrogen, resuspended in 1 mL of 70% EtOH, and centrifuged 20 min at 13,000× *g* at 4 °C. Five hundred µL of the supernatant were collected and incubated with 1.5 mL of reaction mix (1% ninhydrin, 60% acetic acid, and 20% EtOH) for 20 min at 95 °C. The absorbance of samples was measured at 520 nm with a spectrophotometer, and proline concentration was determined by using proline as the standard.

### 4.11. Total Chlorophyll Assay

One hundred mg of *A. thaliana wt*, *strp* mutant, and STRP OE leaves were incubated at 65 °C for 90 min in 5 mL of 80% acetone. Samples were cooled at 25 °C, the supernatant collected, and total chlorophyll (Chl tot), chlorophyll a (Chl a), and chlorophyll b (Chl b) were determined with a spectrophotometer by measuring the absorbance at 663 and 645 nm. Chl concentration (mg/L) was calculated according to [58].

### 4.12. Determination of Hydrogen Peroxide by DAB Staining

In situ H_2_O_2_ production was assessed by staining leaves with DAB, according to [59]. In short, *A. thaliana wt*, *strp* mutant, and *STRP* OE leaves were submerged in a 10 mM DAB pH 6.8, vacuum infiltrated for 15 min, and incubated in the dark for 3 h under stirring. Samples were then bleached to remove chlorophyll by boiling for 15 min in a solution of ethanol: acetic acid: glycerol (3:1:1 *v*/*v*/*v*) as described by [60]. Once cooled at 25 °C, the bleaching solution was removed, and leaves were photographed with a digital camera.

### 4.13. HPLC Analysis of Abscisic Acid

Hormone extraction from *A. thaliana wt*, *strp* mutant, and *STRP* OE plants was performed according to [61]. HPLC analysis was performed on a LC–20 Prominence HPLC system (Shimadzu, Kyoto, Japan) consisting of an LC–20AT quaternary gradient pump, an SPD–M20A photodiode array detector (PDAD), and a SIL–20 AH autosampler (20 µL injection volume). Sample separation was carried out with a Gemini–NX C_18_ column (250 × 4.5 mm, 5 µm particle size) (Phenomenex, Torrance, CA, USA), and hormones eluted with a flow rate of 1.5 mL min^−1^ using a gradient of acetonitrile (ACN) containing 0.1% (*v*/*v*) trifluoroacetic acid in aqueous 0.1% (*v*/*v*) trifluoroacetic acid, at 45 °C. ACN gradient started from 15% and linearly increased to 30 % in 5 min, from 30% to 50% in 5 min, and from 50% to 80% in 2 min, followed by a re-equilibration phase at initial gradient composition in 3 min. ABA detection was performed at 254 nm according to its retention time, UV spectra, and literature data. Purified ABA (Duchefa Biochemie, Haarlem, The Netherlands) was used to build up a calibration curve in the range of 5–200 µg/mL at 254 nm. The reported values indicate the ABA concentration expressed as μg of hormone/g of fresh weight.

### 4.14. Statistical Analysis

Each experiment was performed at least three times using 30 plants for each genotype, considered as a pool for statistical analysis. Statistical significance (* *p* < 0.05, ** *p* < 0.01, *** *p* < 0.001, **** *p* < 0.0001) was assessed by unpaired Student’s *t*-test. All values are expressed as means ± standard error of the mean (s.e.m.).

## Figures and Tables

**Figure 1 plants-12-01704-f001:**
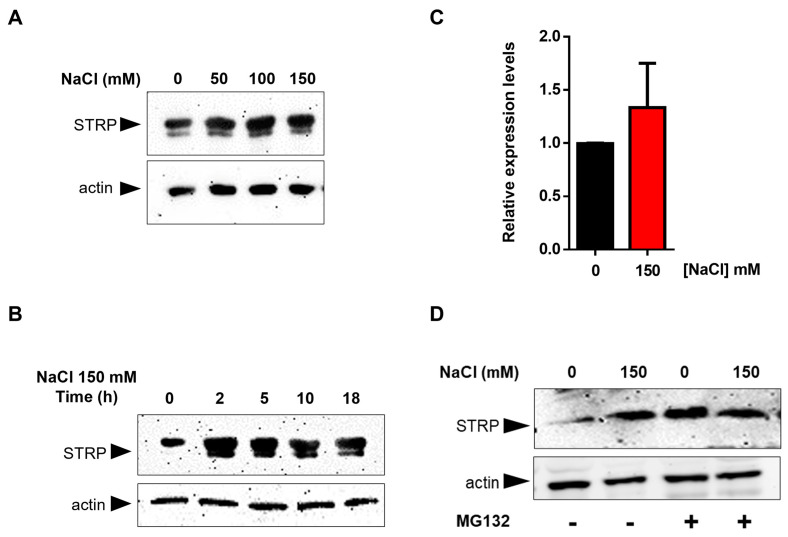
Salt stress increases STRP levels by inhibiting proteasome–mediated degradation of the protein. Salt Tolerance–Related Protein (STRP) levels were assessed by Western blot on two–week–old seedlings treated with 50, 100, and 150 mM NaCl for 18 h (**A**) or with 150 mM NaCl for 2, 5, 10, and 18 h (**B**). Twenty μg of whole cellular extract were separated by SDS–Page, electroblotted on the PVDF membrane, and incubated with the anti-STRP antibodies. Actin was used as a loading control. (**C**) *STRP* expression levels under salt stress were determined by RT-qPCR on total RNA extracted from two-week-old *A. thaliana* seedlings treated with 150 mM NaCl for 18 h. Error bars are s.e.m. of three independent experiments. (**D**) MG132 treatment was performed on the whole cellular extract of two-week-old seedlings treated with 150 mM NaCl for 18 h. Samples (20 μg) were separated by SDS–Page, transferred on the PVDF membrane, and immunodecorated with anti-STRP and anti-actin antibodies as loading control.

**Figure 2 plants-12-01704-f002:**
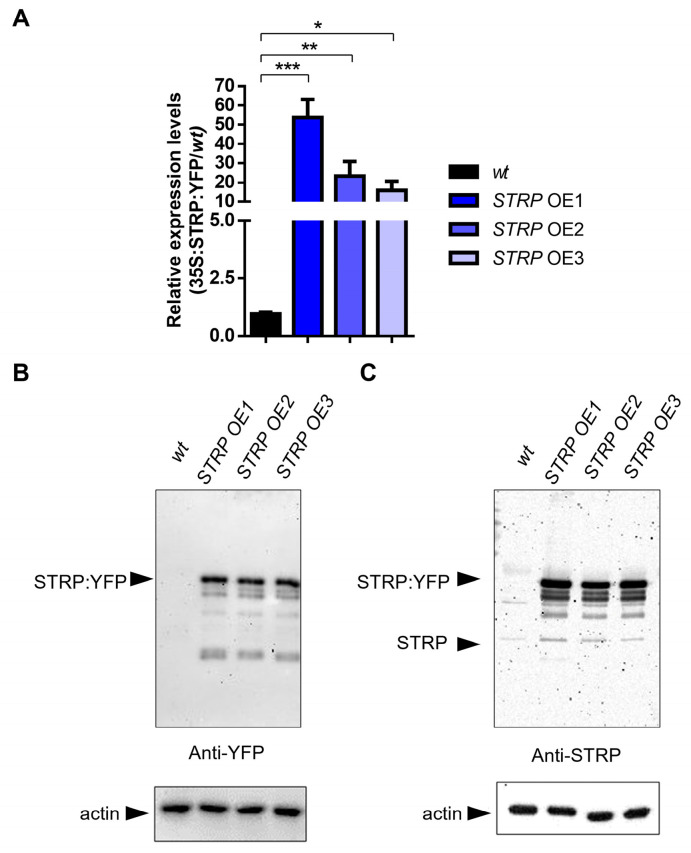
*STRP* overexpression and protein levels in transgenic lines. *STRP* overexpression and protein levels were assessed on *wt, STRP*–overexpressing 1 (OE1), OE2, and OE3 seedlings grown for two weeks on MS medium. (**A**) *STRP* overexpression was determined on total RNA extracted from *A. thaliana*, retrotranscribed, and analyzed by RT-qPCR. Error bars are s.e.m. of three independent experiments. * *p* < 0.05, ** *p* < 0.01, *** *p* < 0.001, by Student’s *t*-test. STRP levels in *wt* plants and the three OE lines were determined by Western blot using 20 μg of whole cellular extract. STRP was detected by incubating the membrane with anti-YFP (**B**) and anti-STRP (**C**) antibodies. Actin was used as a loading control.

**Figure 3 plants-12-01704-f003:**
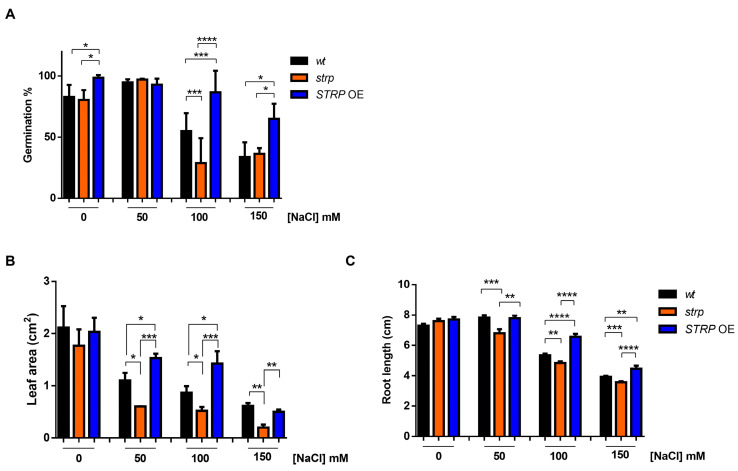
Salt stress affects seed germination, root elongation, and leaf area in *strp* and *STRP* OE plants. (**A**) Germination rate was determined on 200 *A. thaliana wt*, *strp,* and *STRP* OE seeds sowed on Murashige and Skoog (MS) supplemented with 50, 100, and 150 mM NaCl. The seeds’ germination rate was assessed after one week. (**B**) for leaf area measurement and (**C**) primary root elongation, *wt*, *strp,* and *STRP* OE plants were grown for four days on MS medium and then transferred on the same medium supplemented with 50, 100, and 150 mM NaCl. After one-week, leaf area and primary root elongation were measured by analyzing digital pictures with the Easy Leaf Area and ImageJ software, respectively. Error bars are s.e.m. of three independent experiments. * *p* < 0.05, ** *p* < 0.01, *** *p* < 0.001, **** *p* < 0.0001, by Student’s *t*-test.

**Figure 4 plants-12-01704-f004:**
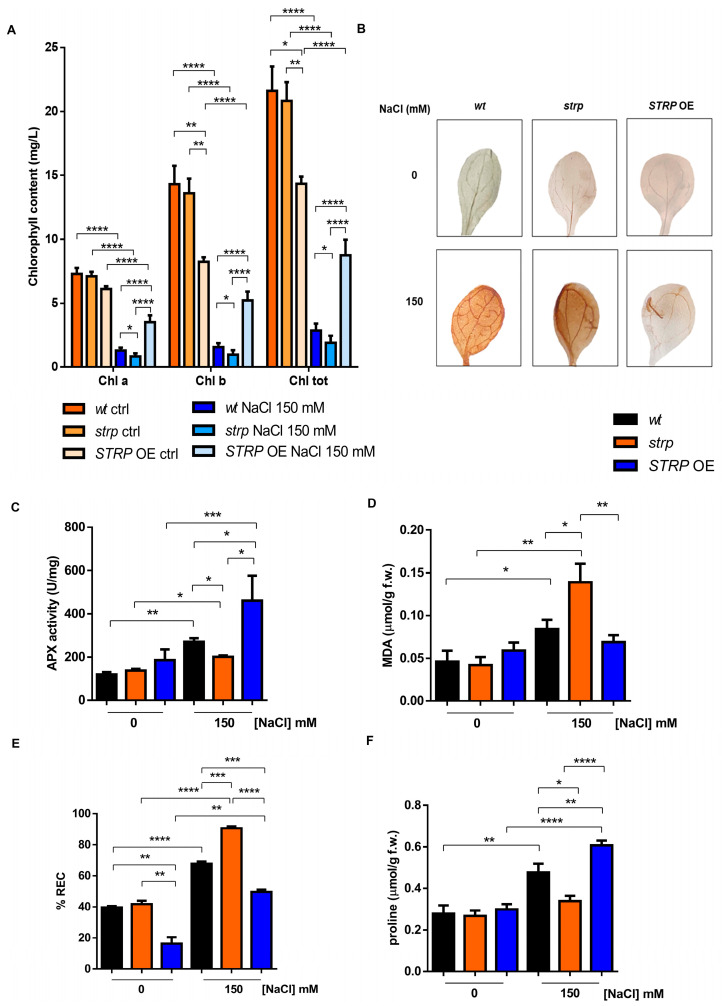
Salt stress differently affects the biochemical responses of *wt, strp,* and *STRP* OE plants. (**A**) Chlorophyll content determined on detached leaves of one-month-old *A. thaliana wt*, *strp,* and *STRP* OE plants treated with a solution 150 mM NaCl every three days for two weeks. Error bars are s.e.m. of three independent experiments. * *p* < 0.05, ** *p* < 0.01, **** *p* < 0.0001, by Student’s *t*-test. (**B**): 3,3′–diaminobenzidine (DAB) staining of detached leaves of two-week-old *A. thaliana wt*, *strp,* and *STRP* OE plants treated with 150 mM NaCl for 18 h. Images are representative of three independent experiments. Ascorbate peroxidase (APX) activity (**C**), malondialdehyde (MDA) (**D**), relative electrolyte conductivity (REC) (**E**), and proline content (**F**) were assessed on two-week-old *A. thaliana wt*, *strp,* and *STRP* OE plants grown on MS medium and then treated with 150 mM NaCl for 18 h. Error bars are s.e.m. of three independent experiments. * *p* < 0.05, ** *p* < 0.01, *** *p* < 0.001, **** *p* < 0.0001, by Student’s *t*-test.

**Figure 5 plants-12-01704-f005:**
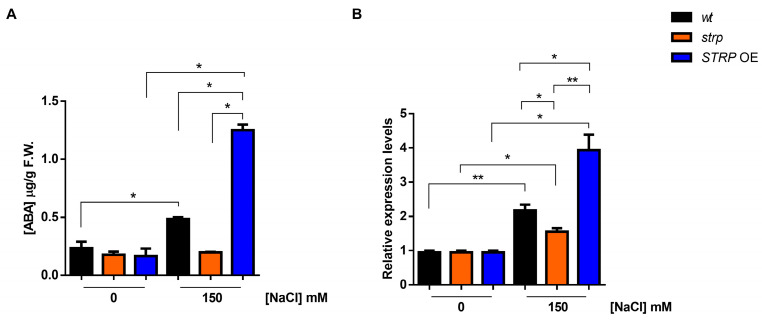
Effect of salt stress on ABA content and *NCED3* expression in *wt, strp,* and STRP OE plants under salt stress. (**A**) HPLC determination of abscisic acid (ABA) content in two-week-old *A. thaliana wt*, *strp,* and *STRP* OE plants treated with 150 mM NaCl for 18 h. (**B**) The 9-cis-epoxycarotenoid dioxygenase (*NCED3*) expression was assessed on total RNA extracted from *A. thaliana wt*, *strp,* and *STRP* OE plants treated with 150 mM NaCl for 18 h by RT-qPCR. Error bars are s.e.m. of three independent experiments. * *p* < 0.05, ** *p* < 0.01, by Student’s *t*-test.

## Data Availability

Data recorded in the current study are available in all tables of the manuscript.

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
