# Peer review of "The Salt Tolerance–Related Protein (STRP) Is a Positive Regulator of the Response to Salt Stress in Arabidopsis thaliana"

_plants, 2023, doi:10.3390/plants12081704_

Round 1

Reviewer 1 Report

The article “The Salt Tolerance-Related Protein (STRP) is a positive regulator of the response to salt stress in Arabidopsis thaliana” claims that STRP performs its protecting functions by reducing the oxidative burst induced by salt stressand plays a role in the osmotic adjustment mechanisms required to preserve cellular homeostasis. also proposing that STRP isa critical component of the response mechanisms to saline stress inA. thaliana. The article is well written and deserve a publication in Plants journal. I just have few suggestions. e.g.,

Abstract:- written well

Keywords:- all keywords should be changed, no key words should be the part of the title.

Introduction:- line 41 and 47 STRP is repeatedly abbreviated, please correct

Introduction is more precise, please add 100-200 words more to explain more about protein, model plant, and its connection with the ABA and salt stress.

Results section is well compiled

Discussion:-line 294-297 results are not well discussed , same is the case in line 303-308, please elaborate more

Material and Methods:- line 372-388, no citations mentioned for these existing methods.

Figure 1-D, STRP-NaCl-0, expression not clear 

Author Response

-Keywords. All keywords have been changed accordingly to the Reviewer’s suggestions.

-Introduction, lines 41 and 47 (original manuscript): the abbreviations have been corrected.

-Introduction: the role of STRP in salt stress responses in Arabidopsis thaliana and the connection with the ABA has been more extensively described.

-Discussion, lines 294-297 and 303-308 (original manuscript): results have been discussed more in detail, and the role of STRP in plant adaptation to salt stress has been better circumstantiated.

-Materials and Methods, lines 372-388 (original manuscript): missing citations have been added.

-Figure 1-D: We agree that the intensity of the band corresponding to STRP is very low. However, the STRP increase in response to salt stress and the effect of MG132 is still clearly noticeable in all the replicates we performed. We hope that it can be considered OK in this form.

Reviewer 2 Report

Manuscript ID: plants-2317666

Title: The Salt Tolerance-Related Protein (STRP) is a positive regulator of the response to salt stress in Arabidopsis thaliana

Authors: Anna Fiorillo, Michela Manai, Sabina Visconti *, Lorenzo Camoni *

Submitted to section: Plant Response to Abiotic Stress and Climate Change

Recommendation: Minor Revision

Considering the major threatened imposed by salinity on agricultural productivity, the present was designed and claimed by Fiorillo et al. is an interesting finding. In crop plants, the salinity adaptation involves a complex processing including alterations in gene expression, hormone signalling regulation and generation of proteins related salt-stress responses. Here, the authors investigated the role of Salt Tolerance-Related Protein (STRP) in salt-induced responses in Arabidopsis thaliana. The result indicated that STRPs rapidly accumulated under salt stress due to a reduction of proteasome-mediated degradation. The Physiological and biochemical responses of the strp mutant and STRP-overexpressing (STRP OE) plants demonstrate that salt stress impairs seed germination and seedling development more markedly in the strp mutant than in wt. The inhibitory effect is also noticeably diminished in STRP OE plants at the same time. Moreover, the strp mutant cannot accumulate the osmocompatible solute proline and does not raise levels of abscisic acid (ABA) in response to salinity stress. It also has a reduced capacity to combat oxidative stress. With STRP OE plants, the opposite result was therefore seen. Overall, the results obtained imply that STRP exerts its protective properties by lowering the oxidative burst brought on by salt stress and participates in the osmotic adjustment mechanisms necessary to maintain cellular homeostasis. According to these findings, A. thaliana's response mechanisms to saline stress include STRP as a crucial component. Overall, the manuscript is good and very suitable for the journal.

1.     The title is eye-catching, interesting and very related to special issue theme of the journal.

2.     This article is written very well and may be helpful for readers especially those who are working in the area of salinity stress/abiotic stress around the globe.

3.     Abstract is good, to-the- point and up to date.

4.     Introduction section is written very well. However, the authors are requested to add some more literature regarding the work done on salt-tolerance related proteins in Arabidopsis thaliana (if possible).

5.     The material and methods section has described in detailed and nicely presented.

6.     Results and discussion are appropriately discussed. The overall manuscript is good. However, there are some errors (grammar, comma, punctuations, spacing etc.) have been observed which is needed to improve before publication of the article. 

7.     I applaud the authors for Figures.

8.     Conclusion section is good but too short. However, I think it needs revision/modifications.

Author Response

-Literature in the Introduction: as requested, a new reference regarding the state-of-the-art on STRP in A. thaliana has been added.

-Result and Discussion: grammar and typing errors have been corrected.

-Discussion: the conclusion has been improved according to the Reviewer’s suggestion.

Reviewer 3 Report

The manuscript entitled ‘The Salt Tolerance-Related Protein (STRP) is a positive regula

tor of the response to salt stress in Arabidopsis thalianais an important paper written well to show the role of STRP in salt stress adaptation using strp mutant and STREP-overexpressing plants. The study includes biochemical, physiological and molecular studies to ascertain the STREP roles. Still, the manuscript needs changes before it is recommended for acceptance.  

- In the title remove full stop in the last.

- Use all abbreviations for the first mention and use them once.

-Avoid use of abbreviations in headings and subheadings.

- The studies for describing the role of STREP should also include name of plant species.

- In Introduction, include more studies which show STREP role in adaptation to other stresses.

- Make sure the use of the term ‘electrolyte leakage’ or REC throughout the manuscript.

- In statistical section, elaborate the statements to show the number of replicates in the independent studies and include the level of significance.

- Last para of discussion, mention the role of STREP in plant adaptation to salt stress through the mechanisms observed in the study.

- Add more literature on salt stress from 2000-2023. 

Author Response

-Title: the full stop at the end of the title has been removed.

-Use of abbreviations: we have used the abbreviations following the journal’s guidelines. They must be repeated at the beginning of each section (Abstract, Main text, Figures). Abbreviations can be used in headings and subheadings.

-Name of plant species: as requested, the name of plant species has been added.

- Literature in the Introduction: as requested, a new reference regarding the role of STRP in other stresses has been added.

- Electrolyte leakage: the terminology has been corrected. Now the only term used throughout the manuscript is Relative Electrolyte Conductivity (REC).

- Statistical section: the number of replicates in the independent studies and the level of significance have been included in the statistical section of the Materials and Methods.

- Discussion: the discussion has been improved according to the Reviewer’s suggestion. In particular, the mechanistic model of STRP in plant adaptation to salt stress has been better explained.

- Literature: as requested, new references on saline stress have been added.